# A three-dimensional view of structural changes caused by deactivation of fluid catalytic cracking catalysts

J. Ihli [1], R.R. Jacob[1], M. Holler[1], M. Guizar-Sicairos[1], A. Diaz [1], J.C. da Silva [1,2], D. Ferreira Sanchez[1], F. Krumeich[3], D. Grolimund[1], M. Taddei [1], W.-C. Cheng[4], Y. Shu[4], A. Menzel [1] & J.A. van Bokhoven[1,3]

Since its commercial introduction three-quarters of a century ago, fluid catalytic cracking has been one of the most important conversion processes in the petroleum industry. In this process, porous composites composed of zeolite and clay crack the heavy fractions in crude oil into transportation fuel and petrochemical feedstocks. Yet, over time the catalytic activity of these composite particles decreases. Here, we report on ptychographic tomography, diffraction, and fluorescence tomography, as well as electron microscopy measurements, which elucidate the structural changes that lead to catalyst deactivation. In combination, these measurements reveal zeolite amorphization and distinct structural changes on the particle exterior as the driving forces behind catalyst deactivation. Amorphization of zeolites, in particular, close to the particle exterior, results in a reduction of catalytic capacity. A concretion of the outermost particle layer into a dense amorphous silica–alumina shell further reduces the mass transport to the active sites within the composite.

[1] Paul Scherrer Institut, 5232 Villigen PSI, Switzerland. [2] European Radiation Synchrotron Facility, 38000 Grenoble, France. [3] ETHzürich, Institute for Chemical and Bioengineering, 8093 Zurich, Switzerland. [4] W.R. Grace Refining Technologies, Columbia, MD 21044, USA. J. Ihli and R.R. Jacob contributed equally to this work. R.R. Jacob is deceased. Correspondence and requests for materials should be addressed to A.M. (email: andreas.menzel@psi.ch) or to J.A.v.B. (email: jeroen.vanbokhoven@chem.ethz.ch)

In the process of fluid catalytic cracking (FCC), porous composites of zeolites and clay are used to transform heavy oil fractions into transportation fuels and petrochemical feedstocks[1]. Over time these composites decrease in catalytic activity. Since this century-old process provides the majority of the world's gasoline[1], there is a constant interest to prolong the lifetime and improve the performance of these composites.

Common FCC catalysts are spherical composites, 50–150 μm in diameter, formed by spray drying. They are composed of ~15–50% rare earth-stabilized (e.g., lanthanum) Y-type zeolite, a crystalline aluminosilicate[2], a functional matrix of calcined kaolinitic clay, an amorphous aluminosilicate, which may contain minor quantities of impurities such as $TiO_2$, and a binder composed of alumina and/or silica[3–7]. This composition allows the composites to withstand operational conditions, while it also allows for the generation of hierarchical porosity. The composites contain an interconnected network of macro- (>50 nm), meso- (2–50 nm), and micropores (<2 nm), which ensures that feed molecules of various sizes experience a retention time in the composite sufficient for conversion into the desired products but brief enough to minimize overcracking[8].

During FCC unit operation, these composites are exposed to rapidly cycling temperatures[9]. They are brought into contact with preheated feed at the bottom of the riser. During their ascent, cracking occurs at about 550 °C on active sites located on the surfaces of the functional matrix and, majorly, the microporous zeolites. A cyclone then isolates the composites from the products and transfers them to a regenerator. In the regenerator coke deposits, i.e., unreacted carbonaceous feed remnants, which cause a transitory deactivation of the catalyst are burned off at about 750 °C. The regenerated composites are returned to the riser, and the process repeats.

The permanent deactivation of FCC catalysts has been under extensive study over the last decades[1]. Previous studies identified persistent coke deposits, the reaction environment in the riser, the hydrothermal conditions imposed by the regenerator, and feed contaminants, including sodium, nickel, calcium, vanadium, and iron[4, 5, 10–14] interacting with and accumulating in the composite as the main reasons behind permanent catalyst deactivation[15–17]. The deactivation of the active zeolites itself through operational wear and impurities is now well considered[18], and zeolites are now designed to account for some material deactivation. However, the loss of zeolites through amorphization during operation, including aspects such as framework collapse and destruction, still remains one of the major expressions of catalyst deactivation. Because catalytic activity is presumed to be directly linked to pore size distributions (PSD), pore surface area, and zeolite accessibility, investigations now focus on changes to the support structure as well. Correspondingly, the pore network architecture is currently a topic of particular interest[1, 15, 16, 19].

Structural changes within single composites, and thus structural contributions to the composites deactivation, can now be imaged by state-of-the-art microscopy[15, 20, 21]. Having visualized an iron-enriched phase exclusively on the exterior surface of deactivated composites, Yaluris et al.[14] suggested these changes to be a result of impurity uptake during operation and to contribute to catalyst deactivation. Based on the glassy appearance of these particles, they hypothesized an impurity-induced melting of binder and matrix components close to the particles exterior to be a reason for deactivation. Such a process would effectively reduce the number of macro- and mesopore openings on the particle exterior. Removal of these so-called diffusion highways, determining the flux of feed molecules to the zeolite domains, ultimately reduces the composite's catalytic activity[22–26]. Martinez et al.[27] extended these observations to other impurities, including nickel and the zeolite-poisoning vanadium and sodium. Structural studies of the pore network[1, 15, 20, 28] further revealed the clogging of diffusion highways by metal-rich impurity deposits[16, 29, 30] and, importantly, the pore network's persistent integrity even after deactivation[31]. Based on these observations, permanent catalyst deactivation is currently postulated to arise in part from the obstruction or removal of diffusion highways close to the particle exterior[15, 16, 32, 33].

Here, we provide an updated view of FCC catalysts and the structural changes that lead to their deactivation. Quantitative ptychographic X-ray computed tomography[21, 34, 35] of a pristine and of two industrial composites obtained from two FCC units operating at increasing severity of catalyst deactivation allows the visualization of differences in porosity and the localization of zeolite, amorphous silica–alumina (ASA), and clay components down to an isotropic resolution of ~35 nm. Independent of deactivation degree, clay/ASA elements are found to be enriched in the outer layer of the composite. In the case of the deactivated composites, we find that this layer, still containing pores and zeolite, condenses and forms a progressively non-porous ASA shell. The formation and growth of this shell thereby effectively isolates the composite's interior from the reaction environment. These observations suggest FCC deactivation to be caused by a combination of zeolite amorphization, which reduces the catalytic capacity of a composite, and the concretion of the outermost composite layer into an isolating shell, which hinders the mass transport into and out of a composite[4, 33].

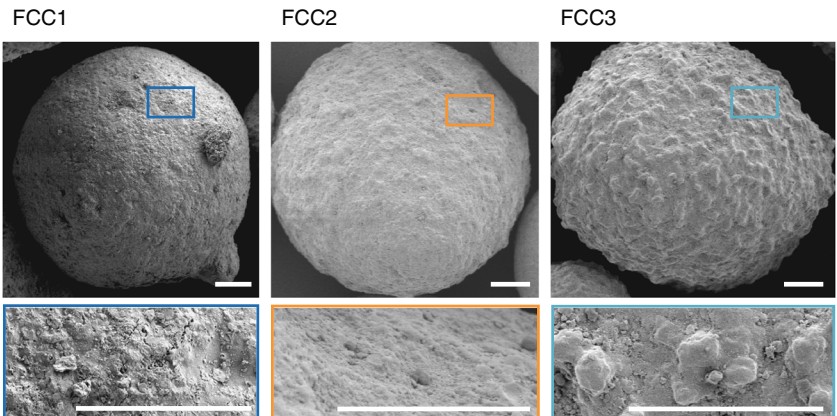

**Fig. 1** Electron micrographs of FCC catalysts. Shown are scanning electron micrographs of an FCC1, FCC2, and FCC3 particle. Highlighted is the morphological transition with catalytic deactivation. *Scale bars* are 10 μm

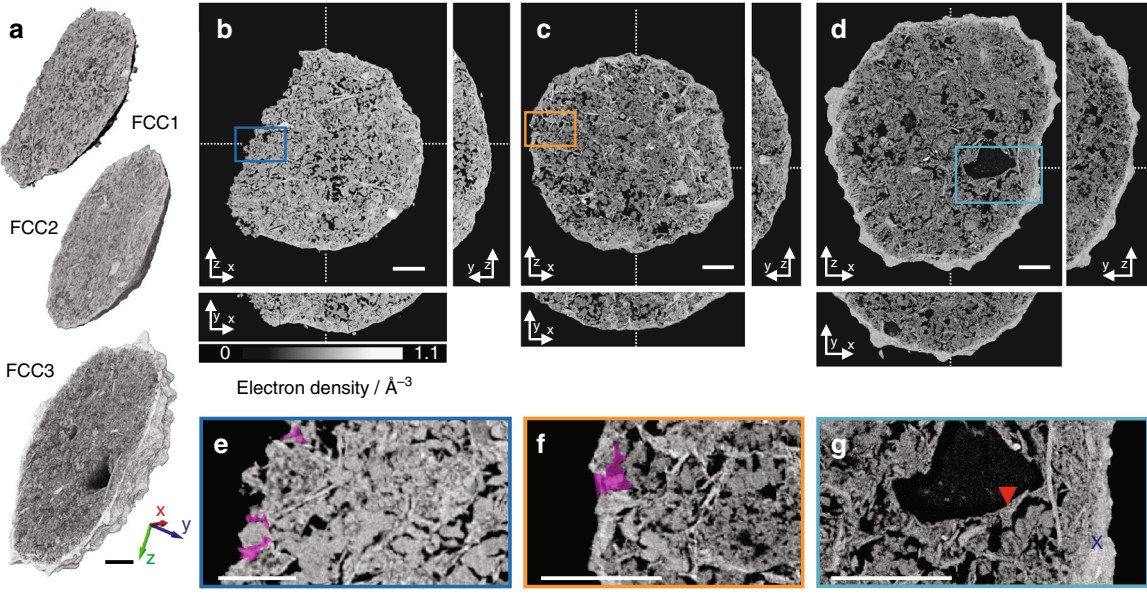

**Fig. 2** Ptychographic image reconstructions. **a** Volume reconstructions of FCC1, FCC2, and FCC3. Orthoslices through the retrieved electron density maps are shown in **b**–**d**, respectively. Presented are *bottom up* (*z*–*x*) and *orthogonal views* (*y*–*z*, *y*–*x*). Cutting planes are represented by *dotted lines*. Shown in **e**–**g** are enlarged versions of selected areas. Common to all subfigures is the linear grey scale for the electron density. Selected diffusion highways (-) are highlighted in *pink*, hydrocarbon deposits by a *red triangle*, and the ASA shell by a *blue cross*. Voxel size is about (20 nm)$^3$. Scale bars are 5 μm

## Results

**Bulk characteristics of fluid catalytic cracking catalysts**. Pristine (FCC1) and commercially deactivated FCC catalysts (FCC2 and FCC3), manufactured under identical conditions, were provided by W. R. Grace Refining Technologies. Deactivated catalysts were retrieved from two industrial FCC units operating at increasing severity of catalyst deactivation. The extracted particles were subjected to a final calcination event. A fraction of these calcined particles extracted from the more destructive unit, FCC3, intended for tomography studies was further subjected to a cracking event in order to observe the state of a FCC particle leaving the riser.

Presented in Fig. 1 are electron micrographs of characteristic FCC particles after synthesis and as extracted from their respective units[36]. Particles of FCC1 measure on average 76 μm in diameter, FCC2 74 μm, and FCC3 91 μm. With progressing deactivation, the initially smooth-surfaced and surface-porous particles undergo a gradual textural transformation and develop a surface comprised of nodules and valleys, resembling surfaces created by localized melt and vitrification events[14]. Compositional analysis by means of inductively coupled plasma emission spectrometry revealed a total alumina concentration of ~50 wt.%. The unit cell of FCC1 was determined to be 2.451 nm, while the unit cell of FCC2 (2.429 nm) and FCC3 (2.430 nm) was virtually undistinguishable (Supplementary Table 1). The FCC3 particles, however, accumulated significant amounts of iron into this new surface morphology (Supplementary Fig. 1). The average iron concentration of FCC1, FCC2, and FCC3 particles is on the basis of $Fe_2O_3$ about 0.5, 0.7, and 1.6 wt. % respectively. Due to the presence of natural iron impurities in clay, around 0.5 wt.% $Fe_2O_3$ is expected for pristine particles[37]. Compared to FCC1, both FCC2 and FCC3 have lower specific surface areas (FCC1 264 m$^2$ g$^{-1}$; FCC2 135 m$^2$ g$^{-1}$; FCC3 118 m$^2$ g$^{-1}$) and a reduced catalytic activity, as measured by conversion values of ~88, ~64, and ~56 wt.%, respectively. The loss of nitrogen-accessible surface area determined via physisorption (Brunauer-Emmett-Teller theory (BET)) is due to a decrease in available zeolite domains (157 to 82 to 50 m$^2$ g$^{-1}$) as for FCC3,

this decrease is slightly offset by an increase in mesoporous surface area (107 to 53 to 68 m$^2$ g$^{-1}$).

**Ptychographic X-ray computed tomography**. To clarify the nature of these morphological and microstructural changes essential to the catalytic activity of FCC particles, we investigated one particle from each ensemble using ptychographic X-ray computed tomography[34], a technique capable of providing highly resolved quantitative 3D density maps[35, 38]. The individual particles were transferred onto tomography pins and without any further sample preparation, we imaged approximately one-third of each particle (Fig. 2). Derived PSD estimates are constrained by image resolution, which was estimated to be better than 31 (FCC1), 44 nm (FCC2), and 35 nm (FCC3) (Supplementary Fig. 2). In comparison with mercury intrusion porosimetry and literature reported BET-derived PSD, such resolution allows us to probe ~90% of the total macro- and mesopore volume (See Supplementary Fig. 3 and Methods for further information).

In the tomograms shown in Fig. 2, the individual composite components can clearly be distinguished. In order of decreasing electron density, these are particulates of titania, lamellar clay, globular zeolite, and a connected network of macro- and mesopores. Two distinct differences between the measured samples are apparent. While the pristine catalyst, FCC1, possesses numerous macropores that directly connect the particle interior with the exterior, the number of such diffusion highways (*shaded pink area* in Fig. 2) is found significantly decreased in the deactivated particle FCC2 or totally absent in FCC3. Instead of these pores, we find the exterior surface of FCC3 covered by an apparently non-porous shell indicated by a *blue cross*. The thickness of this shell is 1.4 ± 0.5 μm. Second, we are able to observe hydrocarbon deposits composed of coke and residual feed, *red triangle*, within the pore space of FCC3.

Plotted in Fig. 3a are electron density histograms of the FCC1 (i), FCC2 (ii), and FCC3 (iii) tomograms in their entirety, alongside histograms of individual composite components such

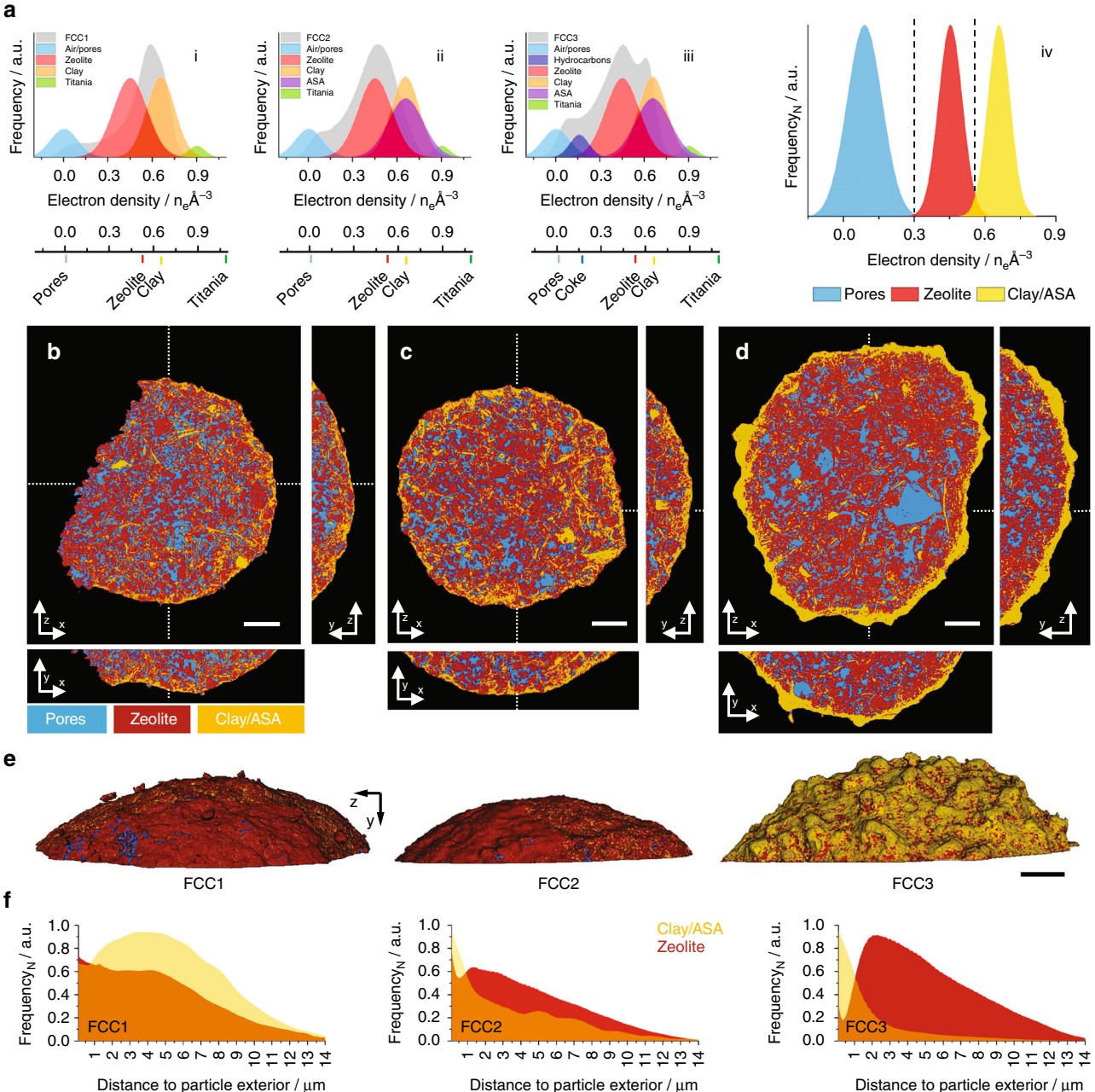

**Fig. 3** Image segmentation of FCC1, FCC2, and FCC3. Shown in **a** are electron density histograms of FCC1 (i), FCC2 (ii), and FCC3 (iii) in their entirety alongside histograms of individual visually pure composite components extracted from FCC3 and calculated electron densities of known catalyst components. Isolated areas with edge lengths of ~250–300 nm taken from single voxel thick slices were chosen for the construction of composite component histograms. This was done to provide an indication of the full electron density spread of a component (i–iii). Shown in (iv) are normalized electron density histograms of the three elementary constituents of a FCC particle, pores, zeolite, and Clay/ASA, created by assigning selected components to one of these constituents. Air and hydrocarbons are grouped as pores (*blue*). Clay, ASA, and titania (TiO$_2$) combined to Clay/ASA (*orange*). Zeolites are shown in *red*. These histograms were obtained from volumes of (250–300 nm)$^3$. Image segmentation of FCC1, FCC2, and FCC3 was initialized using the electron density intersections (---) shown as *dashed lines* in (iv). Resulting segmented tomograms of FCC1, FCC2, and FCC3 are presented in **b**–**d**, respectively. Presented are *bottom up* (z–x) and *orthogonal views* (y–z, y–x). Cutting planes are represented by *dotted lines*. **e** Segmented volume renderings of the exterior of FCC1, FCC2, and FCC3. *Scale bars* are 5 μm. See Supplementary Methods for further information on finite-resolution effects on such tomogram segmentation. Presented in **f** are distance maps of Clay/ASA and zeolite elements with respect to the particle exterior

as zeolite. From Fig. 3a (iii) we observe the calcined clay, essentially an amorphous mixture of alumina and silica, and the iron-enriched ASA shell[14] to possess identical electron densities. Further, a large fraction of zeolites throughout the particle seem to accumulate smaller mesopores with deactivation leading to a shift toward a lighter electron density (Fig. 3a (i and ii); Fig. 2b–d). This shift is a result of partial-volume effects, i.e.,

the mixture of multiple components in a single voxel. The progressing deactivation leads then to the formation of ASA, and the split peak observed in Fig. 3a (iii).

Tomograms were segmented into the three elementary constituents of a FCC particle, pores, zeolite, and clay/ASA, by assigning each identified component to one of these constituents (Fig. 3a (iv)). Because of the colocalization of hydrocarbons and

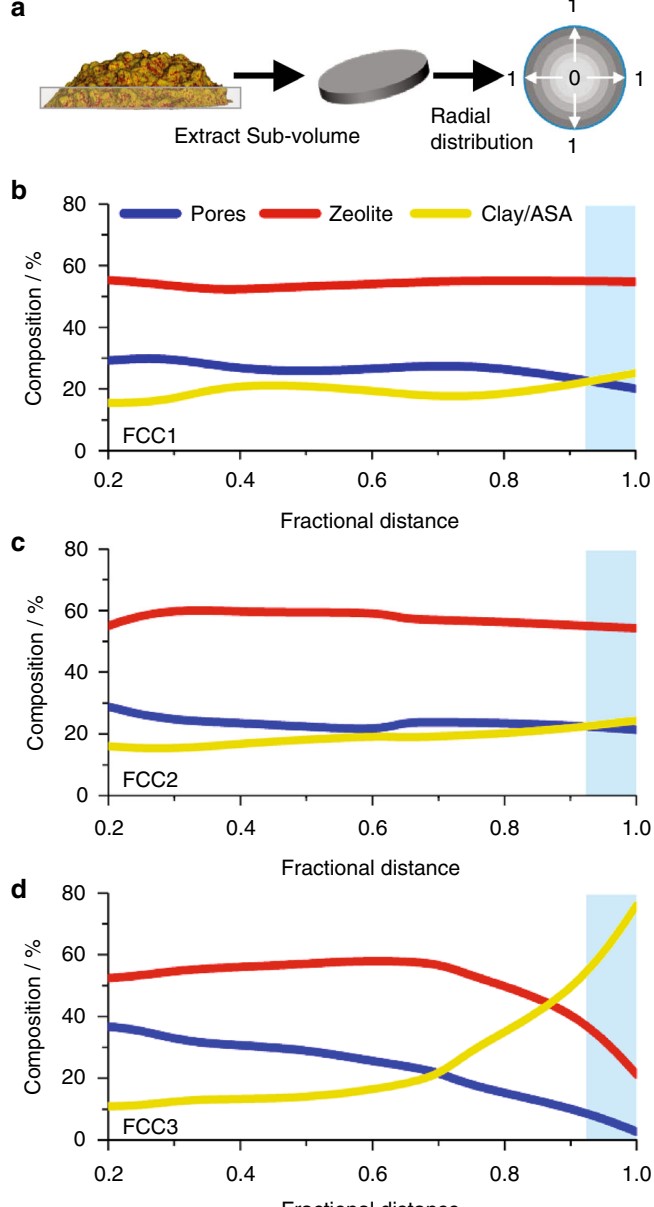

**Fig. 4** Radial distribution profiles of FCC catalyst components. **a** Segmented tomograms (Fig. 3) were cropped along the y-axis to obtain cylindrical subvolumes of the tomogram base (2 μm in height). As a function of fractional distance to the particle exterior, these subvolumes were analyzed in regard to volumetric composition of pores, zeolite, and the grouped clay/ASA fraction. The center of the extracted sub-volume is equal to 0 and the exterior surface is equal to 1. Plotted in **b**–**d** are the volumetric compositions of FCC1, FCC2, and FCC3, respectively. Highlighted in *blue* is a projected shell region ~1.5 μm in thickness

air within the pore network, we assigned air and hydrocarbons to pores. Titania was assigned to clay/ASA, being natural to the composites functional matrix. In general, we observe these three elementary constituents to be well dispersed (Fig. 3b–d). For instance, we determined the intra-particle distances of clay/ASA and zeolite elements to be on average <100 nm. This holds for FCC3 as well, with the notable exceptions of macropores >3 μm$^3$, which we find unique to FCC3 and the dense and increasingly amorphous outer shell. Component segmentation also reveals that this clay/ASA shell is already present in FCC1 and, more so, in FCC2. Both in FCC1 and FCC2, this shell is porous and

contains major amounts of zeolites, which are largely absent in FCC3, see also distance maps of clay/ASA and zeolite (Fig. 3e, f).

The segmented tomograms can further be used to approximate the volumetric composition of each composite (Supplementary Table 1). Overall FCC1, FCC2, and FCC3 possess similar volume fractions of pores, ~23 vol.% of zeolite, ~53 vol.%, and clay, ~24 vol.% (Supplementary Table 1). To assess the spatial distribution of components, we analyzed individual subvolumes, radiating from the core of the composite to the exterior, with regard to their volumetric composition (Fig. 4). Pores, zeolite, and clay/ASA are homogenously distributed throughout FCC1 and FCC2. The outer shell region is composed of ~20 vol.% pores, ~55 vol.% zeolite, and ~25 vol.% clay/ASA. The porosity of FCC3 drops continuously toward the particle exterior (~38 to ~1 vol.%), as does the volume of identified zeolites. The two outermost micrometers in FCC3 are composed of ~77 vol.% ASA and ~21 vol.% zeolite.

In order to establish a connection between individual particle observations and their parent populations, we calculated the specific surface area of detectable macro- and mesopores in the tomograms. The specific surface area for FCC1 is ~132 m$^2$ g$^{-1}$, ~80 m$^2$ g$^{-1}$ for FCC2 and ~92 m$^2$ g$^{-1}$ in the case of FCC3. These specific surface areas exceed by far the BET-determined surface areas of the parent populations, namely 107, 56, and 68 m$^2$ g$^{-1}$. Since, unlike sorption experiments, tomographic observations are independent of access to the reaction environment. This difference can be a result of pores isolated from the reaction environment. Assuming that the tomographically sampled particles provide a good representation of the population average[20], this suggests that independent of deactivation degree up to 30 % of pores are not participating in mass transport.

This conclusion is supported by a detailed analysis of the pore networks in the tomograms of FCC1, FCC2, and FCC3, which reveals the existence of a multitude of minor disjointed networks and one interconnected network accounting for ~80 vol.% of the pores. Shown in Fig. 5a–f are calculated thickness maps[39] of the interconnected pore networks. Here each porous region is fitted with maximal spheres, which fill this region best thereby creating a pore diameter map[39]. Volume reconstructions are presented in the Supplementary Movies 1–3. PSD derived from thickness maps (Fig. 5g) reveal no substantial differences between deactivated samples. Both FCC2 and FCC3 possess a narrow PSD centering at 70 nm. In contrast, FCC1 possesses a much wider PSD whose majority is confined within the range of 30–300 nm. The fraction of pores smaller than 100 nm in diameter is in part magnified by irregular pore surfaces best fitted with a number of small spheres and pore throats connecting individual pockets (Fig. 5d–f)[39]. Comparing the PSD of the pristine sample, FCC1, to the deactivated sample FCC2 and in particular to FCC3, we observed a shift toward larger pores in agreement with the formation of macropores >3 μm detected in Fig. 3.

Plotted in Fig. 5h is the ratio of detectable macro- (>50 nm) and mesopores (<50 nm) as a function of fractional distance to the particle exterior, revealing a homogenous distribution of macro- and mesopores within FCC1 and FCC2, slightly increasing in mesopores when approaching exterior surface for FCC2. In FCC3 we observe a sharp drop in macropores and a respective increase in mesopores as we move to the exterior surface. Figure 5h further highlights the increased fraction of detectable mesopores in FCC1. Considering the equal pore volumes across all tomograms and the increased measureable surface area of FCC1, this iterates the existence of a finer, i.e., a spatially thinner pore network in FCC1.

These pore networks ultimately determine how effectively zeolite domains within the particle are utilized during operation.

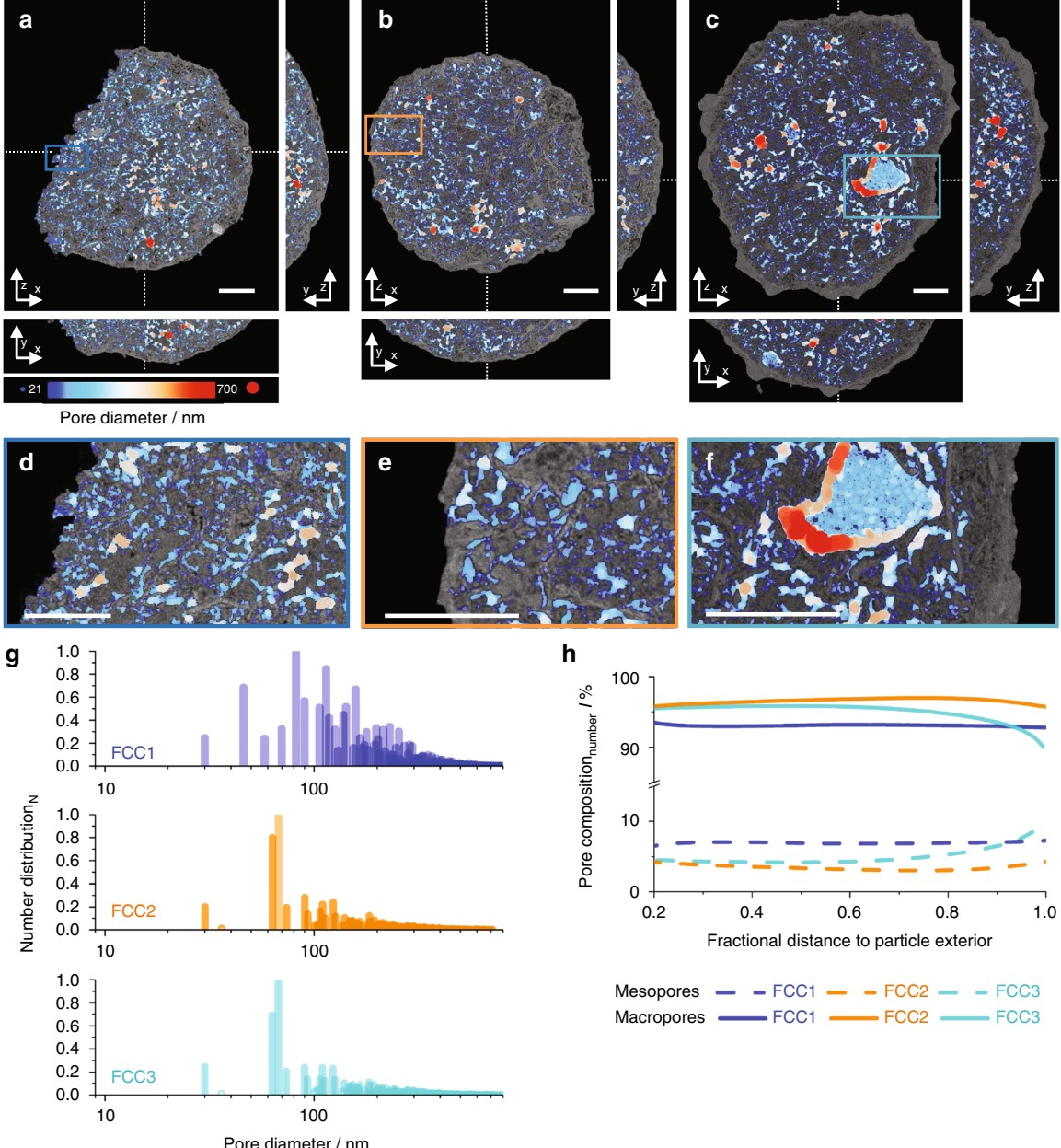

**Fig. 5** Pore network and pore size distribution analysis. Calculated thickness or pore diameter maps of FCC1, FCC2, and FCC3 are presented in **a**–**c**, respectively. Presented are *bottom up* (*z*–*x*) and *orthogonal views* (*y*–*z*, *y*–*x*). Cutting planes are represented by *dotted lines*. Orthoslices are presented in overlay of electron density reconstructions (Fig. 2). The diameter of the fitted spheres is represented by a color map, ranging from 21 (*blue*) to 700 nm (*red*). Shown in **d**–**f** are enlarged versions of selected areas. *Scale bars* are 5 μm. **g** Thickness map-derived pore size distribution (PSD). **h** Radial number distribution of macro- and mesopores in FCC1, FCC2, and FCC3. Subvolumes extracted in Fig. 4 were analyzed in regard to the distribution of detectable meso- (<50 nm in diameter) and macropores (50–800 nm in diameter) as a function of fractional distance to the particle exterior. The smallest spheres considered in **g**, **h** were 31 nm in diameter slightly below the estimated spatial resolution

We therefore aimed to quantify the diffusion highways entrance cross section, for FCC1 we estimated their entrance area to be <5% of the particle exterior surface area. For FCC2 we find only ~1%. Furthermore, the individual entrances rapidly narrow to <150 nm in diameter as we move into the particle. No such connections could be found in the FCC3 tomogram. Rather, we find the pore network to be isolated from the exterior by the dense outer shell on the observable length scale (Supplementary Movies 1–3).

When estimating the theoretical catalytic capacity by the measurable interfacial area of zeolite domains in contact with the reaction environment, including the particle exterior surface and

the pore space but disregarding zeolite micropores and composite elements, which appear isolated because of limited image resolution and field of view, about 70% of the total measureable zeolite surface area can be utilized in FCC1 and circa 60% in both FCC2 and FCC3. Whereas in FCC1 and FCC2 ~10% of this surface area is contributed by zeolites located on the particle exterior, this number is below 2% in FCC3 (Supplementary Table 1).

**X-ray fluorescence and X-ray diffraction tomography**. To query this loss of exterior facing zeolites further, we investigated an

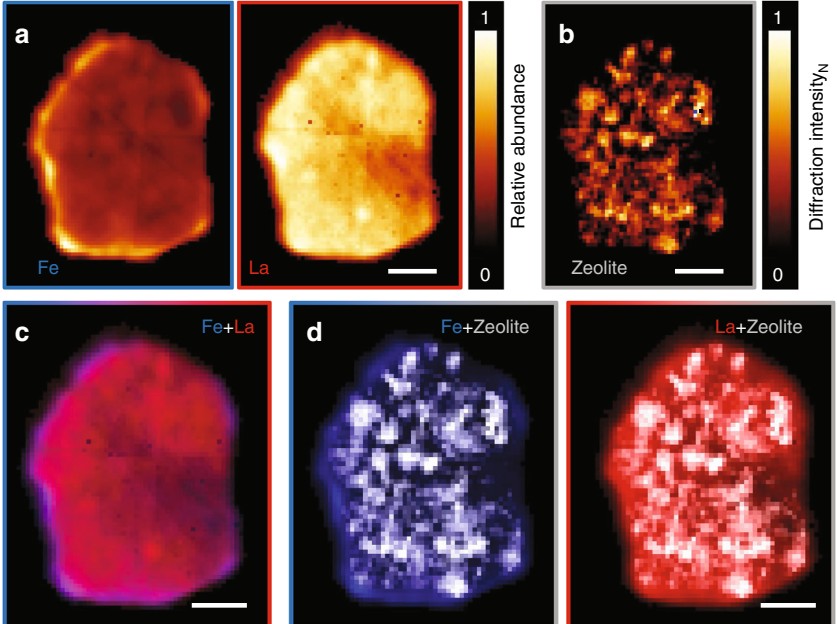

**Fig. 6** X-ray diffraction tomography and tomographic X-ray fluorescence. Presented are central orthoslices through a deactivated coked composite (FCC3). **a** X-ray fluorescence maps of iron and lanthanum. **b** Distribution of zeolite crystals within the composite. **c** Additive combination of iron (*blue*) and lanthanum XRF maps (*red*) highlighting the presence of lanthanum throughout the composite including the iron-enriched shell region, supporting the initial presence of zeolites in this region. **d** Overlay of XRF maps and zeolite crystal distribution. The absence of crystalline zeolite within the shell region in contrast to the presence of lanthanum in this region is indicative of zeolite amorphization. *Scale bars* are 20 μm. Voxel size is (1.5 μm³)

FCC3 particle using combined microbeam X-ray fluorescence (μXRF) and X-ray diffraction tomography (μXRD). XRF (Fig. 6a) reveals the spatial distribution of selected elements, while XRD (Fig. 6b; Supplementary Fig. 4) provides information regarding the distribution of crystalline components and their crystallographic phase. Iron deposits, being representative of the ASA shell (Fig. 3c), are found concentrated at the edge of the particle. Lanthanum, representing the distribution of zeolites in the pristine state of an FCC particle, i.e., before deactivation, is found homogenously distributed throughout the particle[17]. Yet XRD, probing the current location of crystalline components, reveals the confinement of zeolite crystals to the particle interior. By combining both data sets (Fig. 6c, d), we can conclude that the loss of zeolites close to the particle exterior is the result of zeolite amorphization[40]. This is further supported by the Si/Al ratio of clay and ASA compounds (Supplementary Fig. 5). For calcined kaolin clay ($Al_2Si_2O_7$) located within the particle, we observe the expected Si/Al ratio near unity. However, the Si/Al ratio of ASA compounds, i.e., the dense outer shell, is found significantly increased in favor of silicon, a fact explained once we consider the Si/Al ratio (>3) of common hydrocracking catalysts. The intermixing of amorphous zeolites and clay elements into ASA close to the exterior would result in such an observation[17]. And while the degree of zeolite amorphization is most prominent at the edge of the particle, Fig. 6d clearly shows that some zeolite crystallinity within the particle is lost during operation as well.

## Discussion

Examination of the parent populations of pristine and FCC catalysts at increasing severity of deactivation revealed little difference in elemental composition or zeolite unit cell dimension. However, our measurements indicate a significant decrease in specific surface area and functional zeolites. Approximately 50–70% of the active microporous surface area is lost to zeolite amorphization in the deactivated samples. Still, FCC2 and FCC3 particles retained ~73% or ~64%, respectively, of their original catalytic capacity. Our X-ray tomography study of structural changes on the single-particle level provides some insight into the overarching catalyst deactivation behavior.

We observe a shift from the pristine sample toward an initially less electron dense arrangement and, with progressing deactivation, to a subsequently more dense arrangement (Fig. 3a (i–iii)). This trend is in agreement with a progressing zeolite amorphization including zeolite internal mesopore formation, visible throughout the particle (Fig. 2b–d) and ASA development following.

Segmented tomograms reveal a nominally unchanged composition across the examined particles. While a distinct change of the internal structure between the pristine and the deactivated samples is detectable, including a shift toward larger pores and a concurrent decrease in specific surface area, we found no structural changes that lead to a decrease in catalytic activity between the deactivated samples. Rather, we observe the more deactivated sample to possess a slightly increased pore volume, macro- and mesoporous surface area and network connectively, offsetting the effect of progressing zeolite amorphization with regard to catalytic activity. Similarly, the analyzed particles possess near identical interfacial areas of feed-transporting pores with zeolite elements. If such global metrics were applicable, both samples should possess a similar functional capacity[31].

We therefore conclude that the observed drop in catalytic activity is not solely due to the loss of functional zeolites but also a result of localized changes, i.e., to the particle exterior surface. While these changes are not readily detectable by common characterization methods such as Powder-XRD and BET, they are responsible for a drastic reduction of both active sites on the particle exterior surface and accessibility of active sites within the composite.

Others already suggested that macroscopically measurable structural metrics do not accurately describe the diffusion

limitations in or the catalytic performance of FCC particles and that diffusion coefficients, e.g., present a more accurate picture[41, 42]. Wallenstein et al.[36] reported diffusion coefficients in the range $10^{-2}$–$10^{-4}$ cm$^2$ s$^{-1}$, which suggest that intra-particle diffusion in FCC particles occurs predominantly within pores of ~1–80 nm in diameter, the so-called Knudsen regime[43]. Correspondingly, the tomograms presented here register numerous pore throats of the size of voxels within the interconnected pore networks. The major pore network in FCC1, FCC2, and FCC3 was found to contain numerous transport-limiting pore throats, suggesting no significant differences in intra-particle diffusion (Supplementary Movies 1–3). Further, an estimated 1D diffusion length of 140–1400 μm per reaction cycle would indicate that feed molecules have access to all available active sites independent of their distance to the particle exterior.

However, the morphological appearance of the FCC particles changes considerably with increasing deactivation. Particle collisions in the fluidized bed reactor cause smaller particles to fragment and cause surface abrasion[44]. Attrition-created hotspots[45] in combination with operational temperatures are then one reason behind the detected loss and amorphization of zeolites[32] within and on the composite exterior, leading directly to a reduction in catalytic capacity. The amorphization of zeolite crystals close to the exterior surface into an amorphous mixture of silica and alumina combining with adjacent clay elements would then facilitate the observed ASA shell formation. The detection of a shell around FCC2 and to a lesser extent around FCC1 even though porous and zeolite containing points to the catalyst synthesis itself and a gradual formation process occurring over a set number of reaction cycles as possible origins of shell formation[14, 46]. The actual concretion of the shell and the inherent reduction in diffusion highways leads then to a reduction in catalytic activity. Yaluris et al.[14] suggested that feed impurities, mainly iron, introduced during unit operation, are a cause of this process, fostering the formation of low-temperature eutectics.

Yet the loss of zeolite components on the surface has to be highlighted as particularly relevant for catalytic behavior. Not only do these zeolites represent the active components with the shortest distance and biggest cross section to the feed stream, they also constitute a significant fraction of the total amount of zeolite. Assuming a spherical particle with radius of 40 μm, ~10 vol.% is contained within the outermost 1.5 μm.

In summary, a suite of tomographic and microscopic techniques suggest zeolite amorphization and two changes to the particle exterior surface to be driving forces behind FCC deactivation. These changes affect the catalytic activity in two distinct ways. The amorphization of zeolites within and on the particle exterior surface directly results in a reduction of catalytic capacity. The subsequent concretion of this outer, now fully amorphous composite layer into an isolating ASA shell reduces further the mass transport in and out of the composite. These results provide a complementary explanation to the common suggestion that deactivation is governed by physical obstruction of diffusion highways through growing impurity deposits[15, 16]. Importantly, the observed structural changes should be addressable during composite synthesis and unit operation.

## Methods

**Materials**. A pristine and two commercially deactivated FCC catalyst (Ecat) samples, manufactured under identical conditions, were provided by W. R. Grace Refining Technologies. The deactivated FCC particles were extracted from two industrial FCC units. The isolated particles were then calcined to remove any residual coke deposits and separated using a sink/float method[47], which sorts the FCC catalyst population based on differences in skeletal density. Commercial FCC catalysts may contain several additive particles, which are used to control SO$_x$ emission, to improve octane or to reduce sulfur quantities in gasoline. The

sink/float method effectively removes these additives from the cracking catalyst of interest. We imaged one pristine FCC particle (FCC1), one extracted from an industrial unit with a low amount of catalyst deactivation due to iron impurities (FCC2), and one extracted from a unit at the far end of the spectrum (FCC3). All samples had the same starting formulation. They are composed of USY type zeolite, which had been cation-exchanged to 8% lanthanum oxide, kaolin clay, and a silica and alumina binder. The FCC3 particle, which was to be imaged via ptychography, had further been subjected to a cracking experiment. This resulted in a coked sample, simulating the state of an FCC particle leaving the riser/stripper. Chemical composition and selected physical properties are given in Supplementary Table 1. Surface area, mercury intrusion porosimetry, and electron microscopy analysis were conducted on calcined FCC particles to ensure comparable results.

**Electron microscopy**. Scanning electron micrographs were acquired using either a S4500 Hitachi FESEM or with a Quanta Q200 operated at 20 kV. The samples were coated with 5 nm of Pt/Pd. For transmission electron microscopy, FCC particles were first embedded in epoxy resin, and the obtained cylindrical blocks were then polished and cut into fine slices. For scanning transmission electron microscopy (STEM) and energy-dispersive X-ray spectroscopy (EDXS) investigations, these slices were further mechanically thinned and finally ion-milled[48]. High-angle annular dark-field STEM micrographs and EDXS were acquired using either a Hitachi HD 2700CS or with a FEI Talos microscope, both operated at 200 kV.

**Average FCC particle size**. The average particle size of the catalysts was calculated based on electron micrographs. More than 100 particles were measured for each catalyst.

**FCC catalyst composition**. Chemical composition of FCC catalysts was determined by an inductively coupled plasma technique[14].

**Powder-XRD**. Powder-XRD measurements were performed using a Bruker D8 Advanced diffractometer equipped with a CuKα1 X-ray source.

**Surface area determination**. The surface area of FCC catalyst particles was determined by nitrogen physisorption, using a Micromeritics TriStar 3000. Prior to each measurement the sample was degassed at 300 °C for 10 h. The $t$-plot method was used to determine the individual contributions of zeolite and matrix to the total surface area.

**Mercury intrusion porosimetry**. Mercury PSD measurements[20] were carried out using a Micromeritics Autopore IV 9520 unit. The sample was pre-treated at 200 °C for 15 min, then at 540 °C for 1 h. About 0.5 g of the sample was loaded into a 3 ml powder penetrometer. The mercury intrusion pressure was increased in a stepwise manner from 0.1 to 60,000 psi. The volume of mercury intruded after each step is measured and recorded. Acknowledging the unreliability of MIP regarding the analysis of pores with irregular shapes[49], the pore diameter was calculated using the Washburn equation with a contact angle of 140°[50].

**X-ray fluorescence and X-ray diffraction tomography**. Simultaneous microbeam diffraction (μXRD) and microbeam fluorescence (μXRF) tomography experiments were carried out at the microXAS beamline of the Swiss Light Source, Paul Scherrer Institut Villigen, Switzerland. A 17.3 keV incident pencil beam was focused with a Kirkpatrick–Baez (KB) mirror system to a size of $1.5 \times 1.5$ (H $\times$ V) μm$^2$. For X-ray diffraction tomography, diffraction patterns collected from Al$_2$O$_3$ polycrystals were used to calibrate the sample-to-detector distance. The detector, a DECTRIS PILATUS 100 K, was positioned about 5 cm from the sample. Measured diffraction patterns were integrated using the integration software XRDUA[51]. For microbeam X-ray fluorescence, XRF spectra were collected using a Si drift diode detector (KETEK) and fitted using PyMca[52].

To conduct tomography experiments, we glued a single FCC particle on top of a tomography pin using UV-curable resin. After aligning the tomography pin with the sample stage's center of rotation, we simultaneously recorded both XRD patterns and XRF spectra at 91 sample positions with a step size of 1 μm across 181 projection angles, equally spaced by 2°. A total of 16,471 diffraction patterns were collected. The one-dimensional diffraction patterns obtained by Azimuthal integration were then analyzed as a function of position and rotation to construct one sinogram per 2θ value, generating a total of ~5000 sinograms covering a total range of 1.5–56° in 2θ with a step size of ~0.01°. Using these sinograms, a simultaneous iterative reconstruction technique (SIRT) algorithm was applied. Acquired XRF spectra were analyzed in similar fashion, constructing a single sinogram per XRF intensity. This resulted in ~1600 sinograms, covering a total range of 2–18 keV with an energy resolution of ~130 eV. The procedure detailed above allowed us to obtain the full XRF spectrum and the full powder-XRD pattern per voxel of the topographically reconstructed volume[53]. We collected tomographic data for three equally spaced slices, each 1.5 μm in height, across the center of the particle. One of these slices is shown in Fig. 6.

**Ptychographic X-ray computed tomography**. Experimental setup and data acquisition: Ptychographic X-ray computed tomography experiments were carried out at the cSAXS beamline of the Swiss Light Source. For sample-specific information on measurement, reconstruction, and ptychographic tomogram analysis, we refer to the Supplementary Materials. Briefly, using a double-crystal Si(111) monochromator the photon energy was set to 7.3 or 6.2 keV. A horizontal aperture located 22 m upstream of the sample was set to 20 µm in order to illuminate coherently a Fresnel zone plate 150 or 170 µm in diameter with an outermost zone width of 60 nm[54]. Depending on illumination energy, the samples were placed either 1.2 or 0.93 mm downstream of the zone plate's focal plane, resulting in an illumination probe of 3.1–4 µm in diameter at the sample plane. Coherent diffraction patterns were acquired using a PILATUS 2M detector[55] with a 172 µm pixel size, ~7.33 m downstream of the sample. A flight tube was positioned between sample and detector to reduce air scattering and absorption. Measurements were carried out using the positioning instrumentation described in Holler et al.[21, 56] Sampling positions were set using a Fermat spiral scanning grid[57] with an average step size of ~1.2 µm.

Tomography projections were acquired using a binary acquisition strategy as described by Kaestner et al.[58] About 1000–2000 projections of equal angular spacing were acquired across an angular range of 180° for each sample. Each projection was obtained by a ptychographic scan of 253–415 diffraction patterns, each with an exposure time of 0.1 s. The field of view covered in each projection was around ~43 × 15 µm² (H × V). To compensate for the loss of information between individual detector modules, we moved the detector perpendicular to the X-ray propagation direction by a distance slightly larger than the vertical and horizontal module gap widths after two successive projections.

The X-ray dose imparted to the specimens was on the order of $2 \times 10^8$ Gy[59].

Image reconstructions: From each diffraction pattern, a region of 400 × 400 pixels was used in the reconstructions. The resulting pixels are around (21 nm)² in the reconstructed projections. Reconstructions were obtained with a combination of the difference map[60] and maximum likelihood[61, 62] algorithms. Reconstructions were performed by sharing information on the illumination between consecutive scans taken at different projection angles and different detector transverse positions[21, 63]. Prior to tomography reconstructions, the complex-valued projections were aligned and processed as described in Guizar-Sicairos et al.[64] Horizontal alignment was ensured based on tomographic consistency[65]. Tomographic reconstruction of phase projections was performed using a modified filtered back-projection algorithm (FBP)[64]. To mitigate noise in the reconstruction, a Hanning filter was used. The obtained tomograms provide the 3D distribution of the refractive index decrement, $\delta(\mathbf{r})$ which, away from absorption edges, yields directly the electron density[34, 35].

Estimation of spatial resolution: The half-period spatial resolution of ptychographic tomograms was estimated by Fourier shell correlation[66]. The threshold criteria for the Fourier shell correlation was the ½ bit criterion[66]. Fourier shell correlation line plots are shown in Supplementary Fig. 2 yielding resolution estimates for FCC1 to be 31 nm; 44 nm for FCC2, and 35 nm for FCC3.

**Data availability**. The data that support the findings of this study are available from the corresponding authors upon reasonable request.

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

## Acknowledgements

We thank X. Donath and A. Weber for technical support. We also thank the Swiss National Science Foundation (SNF) for the support of the work of R.R.J., J.I. (grant nos. 200021 and 153556) and J.C.d.S. (grant no. 137772). Instrumentation was supported by SNF (R'EQUIP, 145056, OMNY) and the Competence Centre for Materials Science and Technology (CCMX) of the ETH-Board, Switzerland. GRACE Refining Technologies is acknowledged for the provision of samples and material characterization expertise. Electron microscopy was performed at the Scientific Centre for Optical and Electron Microscopy (ScopeM) ETH Zürich. This paper is dedicated to the memory of R. R. Jacob.

## Author contributions

J.I., R.R.J., M.H., M.G.S., A.D., and J.C.d.S. performed ptychographic X-ray computed tomography experiments. J.I., R.R.J., D.F.S., J.I., and D.G. collected and processed μXRF and μXRD data. W.-C.C. and Y.Y.S. conduced EMPA, Hg porosity, and BET experiments. R.R.J., F.K., W.-C.C., and Y.Y.S. performed EM experiments. J.I. analyzed data and wrote the manuscript. A.M. and J.A.v.B. conceived and led the project, gave conceptual advice, and edited the manuscript. All authors read and approved the manuscript.

## Additional information

**Competing interests:** The authors declare no competing financial interests.

