## [Peer Review File · Nature Communications]

Reviewer #1 (Remarks to the Author):

I have re-read this manuscript which was previously under consideration for Nature Materials, but has now been transferred to Nature Communications.

The authors have made some very significant changes to the manuscript, including an analysis of pristine material, which was one of my main requests. This particular change improves the paper by providing a baseline to which the counterpart used materials can now be reliably compared. Overall, I think the authors have done a pretty sound job of responding to the comments of both referees.

I accept and understand the comments provided by the authors about the difficulty of extracting materials and operational parameters for catalysts which have been used under industrial conditions for long time frames. This is a situation which they are unlikely to be able to resolve. This detracts a little bit from the overall impact of paper, but the scientific community still deserves to see this tour-de-force of materials characterisation of these complex and very important FCC catalyst materials.

Hence I am now happy to recommend that this paper should be published in Nature Communications.

Reviewer #2 (Remarks to the Author):

The paper describes changes in material distribution in a petroleum zeolite/clay cracking catalyst. Various techniques are employed to map the distribution of titanium, zeolite, clay and empty pores. A key technique employed is ptychographic tomography. For any one projection, ptychography is known to be very sensitive to the phase change induced in a transmitted X-ray. Back projecting many such images gives a 3D map of the refractive index. This is now a well-established technique. Deriving the volumetric refractive index accurately is a reliable process.

However, to plot out compositional distributions, the next step in the computational analysis is to segment the tomograph according to changes in this single variable (refractive index, which corresponds to electron density). With several material components present, certain assumptions need to be made. Although reference is made to a paper where the calculations of electron density are described [37], any intermixture of material components present on a scale smaller than the voxel resolution limit must make the segmentation ambiguous, at least to a certain extent. All the subsequent results in Figs 3, 4 and 5 rely on the segmentation. The critical diagram is (3) (iv), specifically the positions of the vertical lines.

The authors may well be entirely confident that the boundaries of the segmentation of the three component groups of material type are a very accurate reflection of the actual material distribution. The main trends in 3e are strong. But do any of these plots, or the segmented tomographs from which they derive, depend strongly on the exact segmentation chosen in (3)(iv)?

There is considerable overlap in the probability distribution functions, which themselves depend on the calculated electron density.

In short, I would request the authors to give some indication of how the results depend on the choice of segmentation parameters. First, an estimate of the confidence limits in their calculations of the refractive indices. Second, whether any of the results in the rest of the Figs 3-5, and hence the scientific conclusions they imply, are significantly affected, or even reversed, by any such potential errors.

Paul Scherrer Institut,
WLGA/225
5232 Villigen PSI
Schweiz
Email: Johannes.Ihli@psi.ch

Response to Referees –NCOMMS-17-08408A

We thank the referees and the editorial staff for taking the time to review our revised manuscript, "*Deactivation of Fluid Catalytic Cracking Catalysts, a 3D View of Structural Changes*- NCOMMS-17-08408A". Below you will find the response to the query raised by Reviewer 2. The manuscript has been modified to comply with the reviewer's request as far as possible.

Further changes were made, including: clarifications in the Methods section, the electron density scaling in Figure 3a and 2b to d was corrected, and henceforth scales were unified across images. Lastly, changes to comply with the editorial guidelines of Nature Communications were made. For completeness, the missing EMPA data of FCC1 was added to Supplementary Fig. 1. This information was not yet available at the time of the manuscript transfer. Changes to the manuscript are highlighted in turquoise.

Best regards,

Johannes Ihli

Reviewer #2 (Remarks to the Author):

The paper describes changes in material distribution in a petroleum zeolite/clay cracking catalyst. Various techniques are employed to map the distribution of titanium, zeolite, clay and empty pores. A key technique employed is ptychographic tomography. For any one projection, ptychography is known to be very sensitive to the phase change induced in a transmitted X-ray. Back projecting many such images gives a 3D map of the refractive index. This is now a well-established technique. Deriving the volumetric refractive index accurately is a reliable process.

However, to plot out compositional distributions, the next step in the computational analysis is to segment the tomograph according to changes in this single variable (refractive index, which corresponds to electron density). With several material components present, certain assumptions need to be made. Although reference is made to a paper where the calculations of electron density are described [37], any intermixture of material components present on a scale smaller than the voxel resolution limit must make the segmentation ambiguous, at least to a certain extent. All the subsequent results in Figs 3, 4 and 5 rely on the segmentation. The critical diagram is (3) (iv), specifically the positions of the vertical lines.

The authors may well be entirely confident that the boundaries of the segmentation of the three component groups of material type are a very accurate reflection of the actual material distribution. The main trends in 3e are strong.

But do any of these plots, or the segmented tomographs from which they derive, depend strongly on the exact segmentation chosen in (3)(iv)? There is considerable overlap in the probability distribution functions, which themselves depend on the calculated electron density.

In short, I would request the authors to give some indication of how the results depend on the choice of segmentation parameters. First, an estimate of the confidence limits in their calculations of the refractive indices. Second, whether any of the results in the rest of the Figs 3-5, and hence the scientific conclusions they imply, are significantly affected, or even reversed, by any such potential errors.

Reply to the Reviewer:

The reviewer raises a valid concern, any segmentation based analysis that deals with partial-volume effects, is subject to certain assumptions. The main assumption here, as correctly pointed out, is that the selected segmentation thresholds and method result in an accurate reflection of the actual material distribution.

A comparison of Figure 2 and Figure 3, strongly suggests that such a material distribution is provided by selected segmentation thresholds given the limits in spatial resolution. This notion seems validated by resulting trends and is, in fact, supported by independent secondary methods. In particular, given the respective feature sizes and the minimal overlap in probably distribution (pores vs. zeolite/ clay) conclusions in regard to the detection and identification of the ASA shell, to the loss and amorphization of zeolites, and to major changes in porosity are rather insensitive to partial-volume effects. Thus, there is no reasonable doubt regarding the validity of the main conclusions of the work. Multiple statements are given in the manuscript which stress that results are subject to finite image resolution or partial-volume effects.

The sections on *Matching of Material Phases* and *Segmentation of Material Phases* have been extended and rewritten to provide the reviewer requested information and the limitations of image segmentation as far as possible.

More specifically;

“First, an estimate of the confidence limits in their calculations of the refractive indices.”

The ptychographic tomograms as a result of correlated noise introduced by ptychographic reconstructions do possess an uncertainty in the determination of the mean refractive index, and equivalently electron density. This uncertainty depends on a number of factors; and while we can't provide the reviewer with an absolute uncertainty determination, we can make use of the measured air surrounding the catalyst particle to provide an estimate. The measured air here takes the role of an internal standard of known density. A comparison of the here retrieved electron density with the tabulated density of air we observe an uncertainty between 5 and 10%. This information has been added to the Methods section.

As for the determination of confidence limits and their influence on the selection of segmentation thresholds, these strongly depend on the data points considered for their construction. Gaussians shown in Figure 3a i–iii are based on a limited number of data points, i.e. taken from single voxel thick slices with edge lengths of ~250-300 nm. This extreme was chosen as it visually provides a better indicator of the full electron density spread encountered per component. To determine statically relevant confidence levels we increased the number of data points to a volume of (250-300 nm)³. Such a volume is representative of the average feature size of zeolite and clay/ASA elements following Figure 3a iv has been updated with the corresponding histograms. Notably the overlap in the probability distribution on this feature level is significantly decreased, while the intersection of both materials representing the minimal amount of “cross talk” i.e. the selected threshold level is still evident. The requested histogram derived values and information have been added to the Methods section and Figure 3 caption.

“Matching of Material Phases: Component matching was achieved **in part** by comparing calculated electron densities of known catalyst components with measured electron density histograms of isolated components. Electron densities were **calculated** as described by Diaz et al.¹ **As result of correlated noise introduced by ptychographic reconstructions, the ptychographic tomograms possess an uncertainty in the determination of the electron density. A comparison of the retrieved electron density values of air with the tabulated density of air reveals a spread of 5-10% on single-pixel level.**

The **theoretical electron densities** of hydrocarbon deposits (C_xH_y) were approximated with a density of 1.2 g cm⁻³ and a molecular weight of 18 g mol⁻¹. Those of **Clay and ASA** — being an inter-dispersed amorphous mixture of silica and alumina with unknown composition — were approximated using a mass density of 3.1 g cm⁻³, molecular weight of 80 g mol⁻¹ and atomic number of 29. **Because of its microporous morphology, the retrieved electron density of zeolites is expectedly lower than the calculated electron density. This is a result of partial-volume effects, where a mixture of material components occupies a single voxel.**

To estimate the precision of the tomogram-provided electron densities we first isolated visually pure elements of air, pores, zeolite and clay/ASA that are roughly equal in volume, ~(250-300 nm)³, extracted the corresponding histograms and calculated the respective mean, \bar{x} , standard deviation, σ , and the confidence interval, CI, at the 95% confidence level of these elements. For air these are 0.0002 Å⁻³ \bar{x} | 0.036 σ | 0.0013 CI. The increased spread for pores (0.0209 Å⁻³ \bar{x} | 0.071 σ | 0.0022 CI), zeolite (0.454 Å⁻³ \bar{x} | 0.081 σ | 0.0030 CI) and clay/ASA (0.660 Å⁻³ \bar{x} | 0.055 σ | 0.0018 CI) are attributed to additional component inhomogeneity and partial-volume effects. Data was taken from tomogram FCC3.”

“Second, whether any of the results in the rest of the Figs 3-5, and hence the scientific conclusions they imply, are significantly affected, or even reversed, by any such potential errors.”

The magnitude of potentially miss-classified voxel during segmentation is exceptionally complicated to determine. The former being a function of feature size i.e. the size of individual catalyst components, their distribution and, indeed, their overlap with regards to electron density (overlap in probability distribution histogram). In absence of a validated structural model we can refer to, it is strictly not possible to determine this magnitude on a fundamental level. However we can and already do provide some error estimates.

In the case of pores, by comparing resolution estimates with mercury intrusion porosimetry derived pore size distributions we already show that on limited spatial resolution grounds less than 10 vol.% of pores are on average not correctly classified. Please refer to lines 143 to 148 in the main text and Supplementary Figure 3.

To determine the degree of miss-classification between zeolite and clay/ASA showing the highest overlap in probability distribution no such option exists. To showcase that the selected segmentation threshold between zeolite and clay however provides the most appropriate material distribution we shifted the selected segmentation threshold by half a zeolite standard deviation either towards zeolite or clay. The zeolite standard deviation stated above was chosen as it showed the highest spread. Shown in the figure below are the resulting segmented tomograms. This information has been added to the Methods section.

“Segmentation of Material Phases: Segmentation of ptychographic tomograms allowed the separation of pores, clay/ASA and zeolite components. This was followed by morphological operations to refine the segmentation.² From the intensity histogram, Figure 3, we chose thresholds placed midway between two adjacent peaks as starting thresholds.

To ascertain the correctness of selected thresholds in providing the most accurate material distribution possible, in particular with regards to zeolite and clay/ASA bearing the highest risk of miss-classifying components, we shifted the corresponding segmentation threshold by half a zeolite standard deviation either towards zeolite or clay. The zeolite standard deviation was chosen as it showed the highest spread. Evident was that the general sample characteristics that influence the main conclusions predominantly remain present. In particular, the ASA shell is detected throughout, regardless of segmentation threshold. However, a comparison of the segmented tomograms with the provided electron density tomograms clearly showed that a divergence from the selected segmentation threshold leads to an increasing misidentification of zeolite or clay, or an overrepresentation of these phases, respectively.

Inter-particle distances of clay/ASA and zeolite and distances to the particle exterior were calculated using Euclidean distance maps.”

Shown is the effect a shift in segmentation threshold has on the extracted material distribution of zeolite and clay/ASA. Provided in (a) are the tested segmentation thresholds deviating from the ultimately selected threshold. (b). Shown in the top row are Orthoslices through the retrieved electron density of FCC1, FCC2 and FCC3. Given below are corresponding orthoslices after pure threshold segmentation using the limits provided in (a). The threshold level for pores was kept constant. Pure threshold segmentation was chosen to ensure comparability across tomograms.

Response References

1. Diaz A, Trtik P, Guizar-Sicairos M, Menzel A, Thibault P, Bunk O. Quantitative x-ray phase nanotomography. *Physical Review B* **85**, 020104 (2012).
2. Gonzalez RC, Woods RE. *Digital Image Processing (3rd Edition)*. Prentice-Hall, Inc. (2006).

Reviewer #2 (Remarks to the Author):

I am happy that the authors have comprehensively addressed my questions. In particular, the analysis of the effects of the thresholding has confirmed that the principal scientific conclusions of the paper are sound.

I recommend publication without further alteration.